# Mapping Asia Plants: The Threat Status and Influencing Factors of Rare and Endangered Vascular Plant Species in North Asia (Asian Russia)

**DOI:** 10.3390/plants12152792

**Published:** 2023-07-27

**Authors:** Jianhua Xue, Andrey V. Shcherbakov, Laura M. Kipriyanova, Li Zhu, Keping Ma

**Affiliations:** 1State Key Laboratory of Vegetation and Environmental Change, Institute of Botany, Chinese Academy of Sciences, Beijing 100093, China; 2China National Botanical Garden, Beijing 100093, China; 3Department of the Higher Plants, Biology Faculty, Lomonosov Moscow State University, Moscow 119991, Russia; 4Laboratory of Hydrobiology, Institute for Water and Environmental Problems of the Siberian Branch of the Russian Academy of Sciences, Barnaul 656038, Russia

**Keywords:** mapping Asia plants, plant checklist, rare and endangered vascular plants, North Asian plants, Asian Russia, Red Data Book

## Abstract

In order to effectively protect rare and endangered plants, 27 provincial-level administrative regions in North Asia (the Asian part of Russia) have compiled and published local Red Data Books. In this study, the names (with synonyms) of vascular plants in the 27 provincial Red Books were digitalized and merged into a database of rare and endangered vascular plants in North Asia. The purpose is to reflect the species composition, geographic distribution pattern, and protection level of these plants and their inclusion in the national Russian Red Data Book and the IUCN Red List, and provide a reference for formulating conservation strategies. The dataset has a total of 2079 species, 160 subspecies, and 53 varieties belonging to 667 genera and 143 families. It contains data on 2292 taxa, including family name, genus name, species name and synonyms, protection level, and other information. We also analyzed the main influencing factors, existing problems of rare and endangered vascular plant species, and suggestions for addressing them. We conclude that, to date, the IUCN criteria have not been applied consistently in all regions, leading to an excessive number of species being recorded in the Red Data Books of Asian Russia; specifically, one-third of all floral species are in the regional Red Data Books.

## 1. Introduction

The Red Data Book, an annotated list of rare and endangered or extinct animals, plants, and fungi, is the main document reflecting the survival status of these species. Based on this list, scientific and practical measures for protection, reproduction, and rational use are formulated. Since 1963, the International Union for Conservation of Nature and Natural Resources (IUCN) has maintained the list, also known as the Red List of Threatened Species (https://www.iucnredlist.org/search (accessed on 28 January 2023)).

### 1.1. Background of Red Books in the Study Area

Many countries and regions in the world have compiled and researched their own Red Data Books according to the different endangerment levels of rare and endangered species formulated by the IUCN. With the continuous changes in the ecological environment and the achievements of conservation work, the lists of rare and endangered species are kept in a dynamic updated state [1]. The first botanical part of the Red Data Book of the Union of Soviet Socialist Republics (USSR) was published in 1975 [2] and included about 600 species of vascular plants. This book also contained additional lists of species in need of protection by macro-regions. In particular, 56 additional species were indicated for Siberia and 85 species for the Russian Far East. Since this book was prepared by the Botanical Institute of the USSR Academy of Sciences and the All-Union Botanical Society, it did not have an official state status, in contrast to some regional Red Data Books, which did not have the status of official publications, but were considered popular science and can be equated to Red Lists. The first official Red Data Book of the USSR [3] was published in 1978 by the USSR Ministry of Agriculture (which was then in charge of nature protection issues) and contained 437 species and subspecies of vascular plants. In 1984, the Soviet Union published the second edition of the Red Data Book of the USSR, which included 465 species of vascular plants, including 440 species of angiosperms, 11 species of gymnosperms, and 14 species of ferns [4].

The botanical volume of the first Red Data Book of the Russian Soviet Federative Socialist Republic (as a part of the USSR) [5] contained 465 species of vascular plants. In the post-Soviet period, the Red Data Book of the Russian Federation was officially established by a government decree in 1996, which states that the Red Data Book of the Russian Federation will be maintained by the Ministry of Environmental Protection and Natural Resources based on systematically updated data on the status and distribution of rare and endangered species (subspecies, populations) of wild animals, wild plants, and fungi. In 2008, the botanical volume of the Red Data Book of the Russian Federation was published, which contained 514 species of vascular plants, including 474 species of angiosperms, 14 species of gymnosperms, and 26 species of ferns—more than 4% of the total number of vascular plants in Russia [6].

During the development of the government decree in 1996, Red Data Books were officially established in all regions of Russia. The first official regional Red Books began to appear in the Altai Territory and the Novosibirsk region in 1998 [7,8]. The mass publication of regional Red Books expanded at the beginning of the 21st century. In order to unify these documents, in 2006, the Ministry of Natural Resources of the Russian Federation issued “Methodological recommendations for maintaining the Red Data Book of a constituent entity of the Russian Federation”, No. 02-12-53/5987, dated 27 July 2006 [9]. This document describes in detail the procedure for creating regional Red Books and how they function, the structure of the documentation and individual species descriptions, monitoring works, and the frequency of reprinting. The compilation rules for national and local red lists of the Russian Federation basically comply with the grades and standards of the IUCN Red List of Threatened Species, while at the same time, some regional modifications are made to form individualized evaluation systems [6,10,11,12,13,14,15,16,17,18,19,20,21,22,23,24,25,26,27,28,29,30,31,32,33,34,35,36].

This article provides a comprehensive analysis of the rare and endangered vascular plants included in all 27 Red Data Books at the provincial level in the Asian part of Russia (i.e., North Asia), and all of them are listed in a database that has been created.

### 1.2. Area and Purpose of This Study

North Asia is a vast area of more than 13 million km^2^, or approximately 77% of Russia’s territory. In the west, the border runs mainly along the middle of the Ural Mountains, separating the European and Asian parts of the Russian Federation. The northern and eastern boundaries are bounded by the Arctic and Pacific Oceans. Along the southern border, the region shares borders with Kazakhstan, Mongolia, the People’s Republic of China, and North Korea. North Asia includes the Ural Federal District, the Siberian Federal District, and the Far Eastern Federal District, with a total of 27 provincial administrative regions (Table 1).

North Asia has a complex terrain of mostly mountains and plains. It mainly includes the Ural Mountains, West Siberian Plain, Central Siberian Plateau, mountains of Southern Siberia, East Siberian mountains, and mountains of the Far East. One of the mountain ranges across North Asia (including the Altai, Sayan, and Stanovoy Mountains and the Verkhoyansk–Chukotka folded region) extends from southwest to northeast. In North Asia, there are four major rivers: the Ob, the Lena, the Yenisei, and the Amur Rivers. The climatic characteristics of North Asia vary from coastal oceanic to strongly continental in inland Siberia [37]. There are four main natural zones in latitudinal gradients: tundra, coniferous forests, mixed forests, and steppes (grassland); among them, the coniferous forest zone is the largest. In the Far East, there is a zone of deciduous forests [38]. Due to its high latitude and climatic conditions, North Asia has low plant diversity despite its huge geographic area. The checklist compiled for the whole of North Asia is the “Checklist of the flora of Asian Russia” [39], it contains 6696 species and 265 subspecies from 1187 genera and 191 families.

Although the 27 provincial-level regions in North Asia have compiled and published lists of rare and endangered vascular plants, the information is scattered in certain regions, with a degree of dispersion and locality. There are still some species for which the accepted name is not generally accepted internationally, and there is the issue of synonyms. The purpose of this paper is to integrate the Red Book species lists of 27 provincial distribution areas in Asian Russia and form a distribution map of rare and endangered vascular plants in North Asia, and to reveal the status of threatened vascular plants and the influencing factors. We attempted to compile an integrated checklist of Red Book vascular plants of North Asia in order to contribute to the protection and rational use of these species.

## 2. Results

### 2.1. Checklists of Rare and Endangered Vascular Plant Species of North Asia

We collated 143 families, 667 genera, 2079 species, 160 subspecies, and 53 varieties (a total of 2292 taxa) from the checklists of rare and endangered vascular plant species in 27 regional Red Data Books in North Asia (Appendix A, Figure 1).

The results (a total of 2292 taxa) indicate that in terms of family classification, Fabaceae, Asteraceae, and Ranunculaceae ranked in the top three (Figure 2). In terms of genus classification, *Oxytropis*, *Astragalus* (both in Fabaceae), and *Carex* (Cyperaceae) ranked in the top three (Figure 3).

Among North Asia’s 27 provincial distribution regions, rare and endangered vascular plant species (including subspecies and varieties) are unevenly distributed across different areas. This includes the Yamal-Nenets Autonomous Area, located in the Arctic region of the Western Siberian Plain, which has the lowest number at only 61 taxa. Krasnoyarsk Territory has the highest number with 328 taxa, and the regions with the second-highest numbers are the Republic of Sakha (Yakutia) and Khabarovsk Territory, with 265 taxa (Table 1, Figure 4a).

Among the 27 provincial-level regions in North Asia, the proportion of rare and endangered vascular plants to the total number of vascular plants is between 6% (Republic of Tuva) and 18% (Kurgan Region) (Figure 4b). 

The Red Data Book of the Russian Federation [6] includes 75 families, 176 genera, and 276 species (subspecies and varieties) that are also included in the Red Books of the 27 provincial regions of North Asia, representing 53.7% of the total number on the national Red List of vascular plants (Table 1, Figure 1). We conducted statistical analysis on these 276 taxa, including 40 relict species, 100 species endemic to Russia, and 119 species located at the boundary of the distribution area; among them, the first three categories accounted for approximately 80% of the total. The distribution of the 276 taxa in the 27 provinces and regions in North Asia is also uneven, ranging between 2 taxa (Magadan Region) and 90 taxa (Primorye Territory) (Table 1, Figure 4c).

The comparative analysis results indicate that among all 2292 species (subspecies and varieties) of rare and endangered vascular plants in the 27 provincial regions of North Asia, 65 families, 115 genera, and 190 species (subspecies and varieties) were on the IUCN’s Red List of vascular plants (Table 1, Figure 1), representing 20.9% of all such plants in Russia on the list. The distribution of these 190 taxa in the 27 provinces and regions is also uneven, ranging from 7 taxa (Yamal-Nenets Autonomous Area) to 32 taxa (Republic of Sakha (Yakutia)) (Table 1, Figure 4d).

### 2.2. Comparison of IUCN Criteria with Those in the Red Data Book of the Russian Federation (2008)

We analyzed the similarities and differences in the assessment criteria of threat levels in the three sets of data in this paper. The results show that the evaluation criteria of the Russian national and local red lists conform to the basic framework of the IUCN evaluation criteria, with supplements added and improvements made at the same time (see Appendix A). The IUCN criteria distinguish the following groups: EX (federal level), RE (regional level), EW, CR, EN, VU, NT, and LC. Evaluation is based on quantitative criteria. The IUCN’s EX corresponds to category 0 in the Russian Federation Red Data Book. It can also correspond to category 0 in the regional Red Books, if the taxon disappeared from the territory (this usually refers to taxa found in only 1–3 regions of the country). The EW criterion is absent as a special category in the documentation for the Russian Red Data Book (2008). RE (IUCN) corresponds to category 0 of the regional Red Books, provided that the taxon, having disappeared in this region, has been preserved in other regions of the country. CR (IUCN) corresponds to category 1. EN has no Russian counterpart; it partly consists of taxa in category 1 and partly in category 2. VU corresponds to the Russian category 2. Although there are interval categories in some Red Books, for example, 1–2 or 2–3, such categories are not provided in federal documents at the regional level. NT generally corresponds to the Russian category 3. This category, according to the IUCN criteria, usually includes taxa that, according to one or more quantitative criteria, are close to the boundaries established for classifying taxa in the VU category, but are not yet classified in this category. If, based on an assessment against all IUCN criteria, the taxon is not adjacent to the boundaries for categorizing it as VU, it is categorized as LC. In our practice, these are taxa that should not be included in the Red Book, and consequently do not deserve special protection. When working under the IUCN criteria, category DD includes taxa that cannot be assessed against any of the 12 main criteria, including the subcriteria of some of these criteria, due to a lack of information; that is, both taxa similar to category 4 and those that do not deserve protection are included. The Russian category 5 is absent in the IUCN documents, but it can be classified as LC.

### 2.3. Threat Status of Threatened Vascular Plant Species in North Asia (Asian Russia)

Based on the relationship between the assessment standards of the IUCN and Russia, we determined the endangered level of 2292 threatened taxa in North Asia (Figure 5). Among them, 13 taxa are extinct (accounting for 0.5% of the total), 224 taxa are critically endangered or endangered (accounting for 9.7% of the total), 431 taxa are vulnerable (accounting for 18.8% of the total), and 1530 taxa are near threatened (accounting for 66.7%). Quantitative statistics were calculated for the categories of threatened vascular plants in 27 provincial distribution regions in North Asia (Figure 5). Among them, the proportions of CR and EN species in the Chelyabinsk Region and Primorye Territory are as high as about 50%, while the proportions of CR and EN species in Magadan and Yamal-Nenets Autonomous Area are the lowest at less than 1%.

### 2.4. Influencing Factors of Threatened Vascular Plant Species in North Asia

Based on the descriptions in the Red Book, we conducted incomplete statistics on the influencing factors of the 2292 threatened species, which mainly include 12 types: residential and commercial development, agriculture and aquaculture, energy production and mining, transportation and service corridors, biological resource use, human intrusion and disturbance, natural system modifications, plant diseases and insect pests, geological events, climate change and severe weather, and low ability of species to reproduce and spread, with narrow distribution. We counted a total of 6679 items of risk factor data, and their proportions are shown in Figure 6. Among them, the share of external risk factors, such as human activities and natural disasters, is about 66%, while the share of risk factors caused by the inherent biological characteristics and ecological habits of species is about 32% (Figure 6).

## 3. Discussion

### 3.1. Quantity and Geographic Distribution of Rare and Endangered Vascular Plants in North Asia

The local Red Data Book for North Asia’s 27 provincial administrative regions includes 143 families, 667 genera, and 2292 species (subspecies and varieties) (Table 1, Figure 1), with 6961 species (subspecies) from 1187 genera and 191 families [39], comprising 32.9% of the total on North Asia’s checklist of vascular plants. Therefore, about one-third of North Asia’s vascular plants are included under different classes of protection.

Based on the three sets of data in this paper, these numbers basically reflect the geographical distribution pattern of rare and endangered vascular plants in North Asia (Figure 4a). It can be seen that the regions with the largest number of vascular plants are the Republic of Sakha (Yakutia) (9; Figure 4a) and Krasnoyarsk Territory (13; Figure 4a), because they have the largest area. After removing the influence of factors on the total area, it can be seen that the regions with the largest distribution of rare and endangered vascular plants are Primorye Territory, Sakhalin Region, Kamchatka Territory, Khabarovsk Territory, Republic of Altai, Altai Territory, and Chelyabinsk Region (Figure 4a,c,d). These regions are located in mountainous areas with high habitat heterogeneity at low latitudes; the first four regions benefit from the influence of marine climate conditions, which are more conducive to biological survival and evolution, and the existing species diversity is high [38]. Taking the Primorye Territory as an example, 90 taxa (accounting for 42% of the total threatened species) were listed in the Red Book of the Russian Federation (2008), with 11 endemic species and 17 relict plants located, and 58 taxa at the boundary of the distribution area, accounting for 80% of the total. In addition, the environment in these areas is also suitable for human habitation, with large population distribution and density and relatively large human disturbance, which may be the reason for the large amount of rare and endangered plants.

### 3.2. Influencing Factors of Rare and Endangered Vascular Plants Species in North Asia

Based on the statistical results of the influencing factors of threatened species, it is evident that the most important factors are human economic activity, livestock farming and ranching, tourism and recreation, gathering of terrestrial plants, logging and wood harvesting, mining and quarrying, roads and railroads, climate change and severe weather, and fire and fire suppression. For example, in the Trans-Urals, the abundance of *Stipa pulcherrima* K. Koch has decreased due to pasture cultivation, and the species may be endangered or disappear [30,33,35]. In addition, the limiting factors of the species themselves, such as low reproductive and population dispersal ability, narrow ecological range, and low population numbers, also lead to species endangerment (Figure 6). For example, the competitiveness of *Eutrema cordifolium* Turcz. ex Ledeb. [20,21,22] and *Micranthes brachypetala* (Malyschev) Tkach [13,17,18] is weak and their survival ability is reduced.

### 3.3. Problems in the Red Lists of 27 Administrative Regions in North Asia and Suggestions for Addressing Them

At the level of the 27 regions in North Asia, the number of Red Book species (subspecies and varieties) accounts for 6% (Tuva Republic) to 18% (Kurgan Region) of all vascular plants in the region (Figure 4b). However, combining the 27 provincial-level Red Book lists, the number of species (subspecies and varieties) accounts for 32% of the total number of vascular plants in North Asia. There are several reasons for such a huge disproportion, with one-third of North Asia’s vascular plants being included under different classes of protection in the Red Data Books of Asian Russia.

#### 3.3.1. Inclusion of Species at the Borders of Ranges

The vast territory of Asian Russia, from both west to east and north to south, includes the limits of distribution of many species. The populations of many plant species that are on the edge of their range are often less resilient than those in the center of the range. It was shown for aquatic species that growing at the limits of their global range was the most frequent reason for their inclusion in regional Red Data Books of Asian Russia, and 61 species represent 64% of the list [40]. Thus, the Red Book of the Magadan region [12] includes species with wide distribution such as *Ceratophyllum demersum* L. and *Potamogeton pectinatus* L. (*Stuckenia pectinata* (L.) Börner), and the Red Book of the Irkutsk region includes *Hydrocharis morsus-ranae L. and Sagittaria sagittifolia L.,* which are on the edge of their range. Of the total 224 vascular plants in the Amur region Red Book, 81 are on the edge of the distribution area or ecological distribution area, accounting for 36% of the total.

#### 3.3.2. Confusion between Concepts of “Rare” Species and “Endangered” Species

A large number of taxa are listed in the regional books of the Asian part of Russia because some of our colleagues confuse the concepts of “rare” species and “endangered” species. Regarding the former, the rarity of a taxon is often associated with natural causes or their ecological and biological features. *Triglochin maritima* L. and *Spirodela polyrhiza* (L.) Schleid. are included in the Red Data Book of Kamchatka Territory [10], and *Potamogeton perfoliatus* L. in the Red Data Book of Chukotka Autonomous Area [11]. Some plants are too small in size and are often overlooked by researchers. Inconspicuous species such as representatives of the genus *Elatine*, *Coleanthus subtilis* (Tratt.) Seidel ex Roem. et Schult., and *Ranunculus reptans* L. are often included in Red Data Books, although they are often not threatened, as they are widespread. These species, as a rule, have no commercial value and are not collected by local people. In this case, conservation management decisions will hardly affect the abundance of the taxon in nature. In the case of a threatened taxon, as a rule, we are looking at the action of anthropogenic factors, which can be weakened or removed by making managerial decisions.

#### 3.3.3. Lack or Misuse of Population Trend Information

It was shown that most of the aquatic plants in the regions of Asian Russia are not considered to be threatened; approximately one-half have stable populations, there are no data for one-third of the species, and one-fifth of the species are increasing, while most of the protected species in the region listed in the Red Data Book of Russia are declining (94%) [40]. Most of the seriously declining aquatic species occur in West Siberia and the Far East, areas with the largest human populations and highly developed industry, including oil and gas production in West Siberia and mining of gold, diamonds, and complex ores in the Far East, as well as intensive agriculture in the south of both areas [41].

#### 3.3.4. Inclusion of Species under the Control of Natural Factors

Some Red Books list species whose distribution is regulated by purely natural factors (primarily geomorphology, climate, or hydrology). As an example, in the Republic of Altai [25], with a predominantly mountainous terrain where there are few reservoirs of mesotrophic status, *Hydrilla verticillata* (L. f.) Royle, which is commonly found in neighboring plain regions, is among the rare species in the Red Book. Some species have a wide distribution, but are rare because of their association with cold oligotrophic lakes (such as *Isoëtes* spp., *Sparganium angustifolium* Michx., and *Subularia aquatica* L.).

A notable effect of climate change is the northward range expansion of both widespread native species and non-native species, which has already been recorded for some aquatic plants and many southern species now occurring north of their conventional ranges [40]. It is inappropriate to include such plants in the Red Books. If natural conditions change in a favorable direction for them, they will expand their range and numbers; if unfavorable, no single organizational measure will save them [41].

#### 3.3.5. Inclusion of Species Characteristics at Individual Stages of Restorative Succession

It is also inappropriate to include such species in the regional Red Data Books [41]. Usually, they have no economic value. The organization of natural areas for their protection only accelerates the process of loss of such plants from the flora, since they require regular violation of the integrity of the vegetation cover for their normal existence.

#### 3.3.6. Study of the Territory

As of 2020, six regions of Asian Russia did not have up-to-date floristic reports; in 11 regions, these reports were presented by key manuals in which the keys for identifying plants take up a lot of space, and only 10 had full-fledged reports (flora or synopsis of flora) [42]. However, an analysis of the regional Red Data Books in Central Russia showed that their quality was correlated with the quality of the floristic knowledge of the territory [43]. An increase in the number of aquatic plants (from 4 to 11) in consecutive editions of the Red Data Books of the Novosibirsk region [8,28] indicated that surveys of vegetation took place between those publications.

#### 3.3.7. Questions about Synonyms

After combining the vascular plant lists in the Red Books of 27 provincial distribution areas and removing duplicate data, a total of 2440 records were obtained. When these records are matched against the Catalogue of Life, which includes 495 taxa synonyms, it can be seen that the proportion of synonyms in the data is as high as 22.0%. This shows that the problem of synonyms is widespread.

#### 3.3.8. Classification of Species Protection Classes

Among the 27 provincial distribution areas in North Asia, the years of publication are different, as well as the standards for division. In most regions, the Red Books are compiled according to the requirements of the “Methodological recommendations for maintaining the Red Data Book of a constituent entity of the Russian Federation”. The assessment of taxa protection levels basically refers to the relevant assessment standards of the International Union for Conservation of Nature (IUCN) and the Red Data Book of the Russian Federation [6]. According to these recommendations, regional Red Books must be updated every 10 years, but, as indicated by Table 1, this requirement is not always met. Although these recommendations allow the use of rarity categories that differ from those in the Red Data Book of the Russian Federation, in practice, it is almost impossible to do this.

First, descriptions should indicate the conservation status of the taxon in all neighboring regions. In this case, if an author uses other categories (including those recommended by IUCN), the data for the region become incomparable with the data for neighboring regions. And since the Red Books in neighboring regions are usually published in different years, we do not know how this contradiction could be resolved.

Second, as we pointed out above, the flora of many regions of Russia is still poorly studied. Not all regions of the country have a sufficient number of qualified botanists. As shown by a seminar on the use of IUCN criteria in the Russian Federation at the regional level held in Moscow in the early 2000s, the floristic information available for most regions of the country does not allow a qualitative determination of the categories of protection for many plant taxa. There are Russian studies on the current situation with rarity categories and recommendations for changing it [44,45,46,47].

#### 3.3.9. Organizational Problems

Today, we can state that the prepared and approved letter of the Ministry of Natural Resources of the Russian Federation No. 02-12-53/5987 dated 27 July 2006, “Methodological recommendations for maintaining the Red Book of a constituent entity of the Russian Federation” [9], did not solve the problem of scientific and methodological unification of the preparation of regional Red Books. Only the problem of normative–legal unification was solved. The content of regional Red Books is largely determined not by the rules, but by the expert opinions of their compilers. For this reason, books on different subjects of the Russian Federation, and even their publication at different times on separate subjects, cannot be compared, since their differences will be determined by the subjectivity of the compilers. Experts arbitrarily assign IUCN threat categories without assessing regional populations, since in most regions, the Ministry of Natural Resources does not fund expeditionary work to study populations of rare species. In the absence of rules, the importance of preparing printed publications is growing, while the importance of collecting special information about rare organisms is decreasing. The beneficiaries of the existing system are both officials and compilers. For the former, published books are important as reports for higher authorities and as gifts for their colleagues. The benefit to compilers, on the other hand, is that they can obtain material benefits (by writing the maximum number of essays) at a low cost of research. That is why the number of unsubstantiated reports is growing and the abundance of books is increasing [48].

Thus, we summarized the information about compiling red lists in the vast territory of Asian Russia, which surpasses Canada, China, and all of Europe in size. Russia has a long way to go, starting from understanding the importance of and providing state support for the branch of knowledge known as conservation biology in order to study population dynamics and threats. It is necessary to rebuild the entire system of compiling red lists of species in need of protection. Undoubtedly, it is necessary to adopt the vast global experience gained by the IUCN in compiling the Red Lists (see, e.g., [49,50,51,52,53,54]). Currently, IUCN criteria are not applied consistently in all regions, leading to an excessive number of species in the Red Data Books of Asian Russia; specifically, one-third of all floral species are in the regional Red Books, the collection of and damage to which, according to a recently adopted state law, are subject to criminal liability. Legislative acts are also needed to abandon expensive printed Red Data Books and instead publish dynamic, easily updated electronic lists that can reflect the real problems of species conservation. There will also be objective difficulties associated with the disproportions in the size of the territories in the regions of Asian Russia and the number of qualified and interested researchers.

Asia is home to many vastly diverse and environmentally sensitive countries, and it is vital for these countries to develop practical conservation strategies based on red list data [55]. As we have shown, in the Asian part of Russia, there is a lot of work to be done to unify and rationalize the compilation of the red lists themselves.

## 4. Materials and Methods

### 4.1. Data Sources

We followed the administrative divisions of the Asian part of Russia comprising 27 first-level regions: 12 regions (oblasts), six territories (krais), five republics, three autonomous areas (autonomous okrugs), and one autonomous region (autonomous oblast) (Table 1). The names of the Russian regions are taken in accordance with the English version of the Constitution of the Russian Federation (http://www.constitution.ru/en/10003000-04.htm (accessed on 19 November 2022)). We collected and digitalized the vascular plant data in the Red Data Book of the Russian Federation (514 species (subspecies and varieties) altogether) [6] and the local Red Data Books for the 27 provincial administrative regions (Table 1) [13,14,15,16,17,18,19,20,21,22,23,24,25,26,27,28,29,30,31,32,33,34,35,36,37,38,39]. We also downloaded the IUCN Red List of Russia’s 908 species (subspecies and varieties) of vascular plants from the IUCN website (https://www.iucnredlist.org/search (accessed on 28 January 2023)).

### 4.2. Construction of Database of Threatened Vascular Plant in North Asia

This dataset includes the rare and endangered vascular plants listed in 27 provincial-level Red Books in North Asia (published or released), with a data volume of 2292 items (see Appendix A). The data include information such as family name, genus name, species name, synonyms, protection category, and distribution area (Appendix A).

To harmonize the taxonomy and nomenclature applied in different data sources, we followed the taxonomy of the 2019 Annual Checklist of the Catalogue of Life (http://www.sp2000.org.cn/names_match_services (accessed on 2 November 2022)). The spelling of names and authors in the Catalogue of Life was accepted as well. A few names that were missing in this global index of species were nevertheless included in the checklist. In this dataset, the classification is based on PPGI [56] for lycophytes and ferns, for gymnosperms [57], and APGIV [58] for angiosperms. The family, genus, Latin name, important synonyms, and distribution regions in Asian Russia were recorded for each vascular species. The synonyms were taken from accepted names in the data sources covered by our compilation. In addition, in this database, if the same threatened species was included in multiple provincial regions at different levels, we chose the one with the lowest level as the target.

### 4.3. Statistical Analysis of Data

Based on the Red List of vascular plants in 27 regions of North Asia, we calculated the numbers and distribution patterns from the Red Data Book of the Russian Federation and the IUCN Red List. We analyzed the similarities and differences in the assessment criteria for threat levels in the three sets of data in this paper, and selected a unified IUCN standard for merging and other comparative analyses. We also attempted to summarize and analyze the factors influencing these threatened species.

## 5. Conclusions

Our research shows a huge disproportion, with one-third of North Asia’s vascular plants being included under different classes of protection in the regional Red Data Books of Asian Russia. One of the common shortcomings of a significant number of regional Red Data Books is the inclusion of many species that are not actually threatened by extinction and do not need special protection measures.

The most significant problems of the regional Red Books of Asian Russia, according to the authors, are confusion between the concepts of “rare” species and “endangered” species, the frequent inclusion of species at the borders of ranges, the inclusion of species under the control of natural factors, the inclusion of species characteristic of individual stages of restorative succession, insufficient knowledge of the territory and a lack of data on population dynamics, and organizational problems associated with maintaining the books.

It is necessary to change the situation, and a more critical approach is needed for the selection of species to include in Red Data Books. The main criteria for including taxa in Red Books should be that their existence is threatened as a result of human activities and that we have the ability to influence their process of extinction.

## Figures and Tables

**Figure 1 plants-12-02792-f001:**
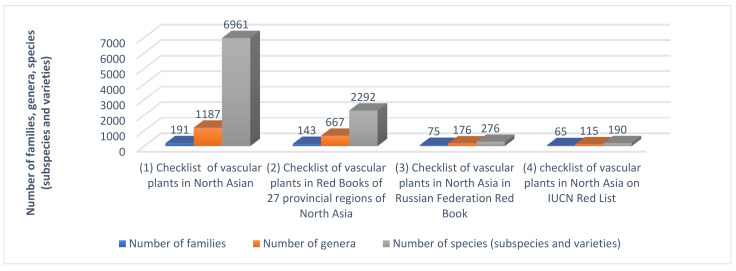
Quantitative statistics on numbers of families, genera, and species (subspecies and varieties) included in the (1) checklist of vascular plants in North Asia [39], (2) checklist of vascular plants in Red Books of 27 provincial regions of North Asia, (3) checklist of vascular plants in North Asia in Russian Federation Red Data Book, and (4) checklist of vascular plants in North Asia on IUCN Red List.

**Figure 2 plants-12-02792-f002:**
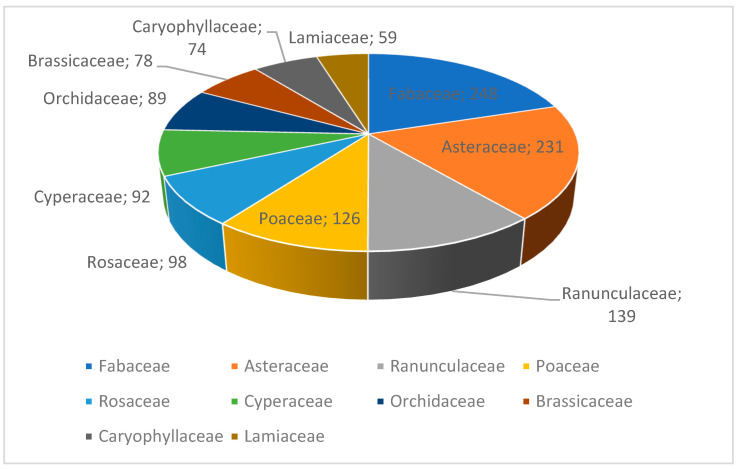
Quantitative statistics for top 10 families and their percentages among total number of rare and endangered vascular plants in North Asia, from highest to lowest.

**Figure 3 plants-12-02792-f003:**
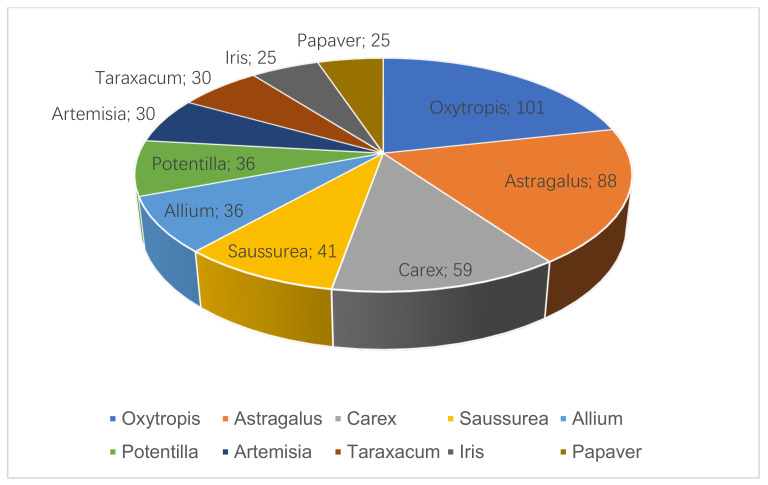
Statistics for top 10 genera and their percentages among total number of rare and endangered vascular plants in North Asia, from highest to lowest.

**Figure 4 plants-12-02792-f004:**
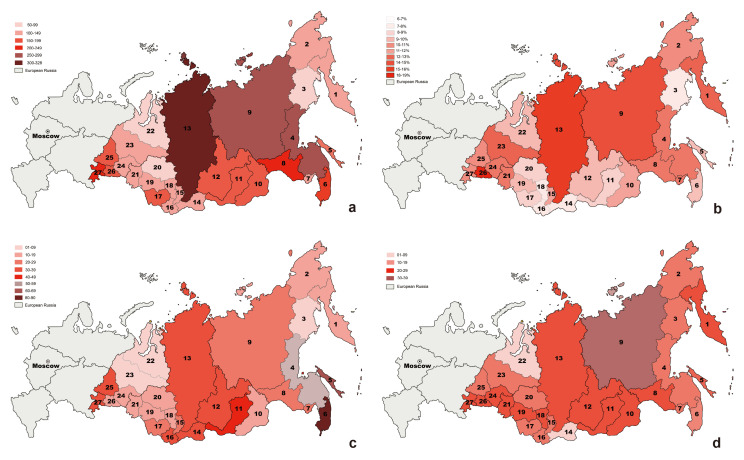
(**a**) Administrative regions and numbers of rare and endangered vascular plant species (subspecies and varieties) in Red Data Books of 27 provincial regions in North Asia. (**b**) Proportion of rare and endangered vascular plants to total number of vascular plants in regions of North Asia. (**c**) Quantity distribution map of vascular plant species (subspecies, varieties) in 27 regions of North Asia in the Red Data Book of the Russian Federation [6]. (**d**) Quantitative distribution of vascular plants species (subspecies, varieties) on IUCN Red List in 27 regions in North Asia. 1—Kamchatka Territory; 2—Chukotka Autonomous Area; 3—Magadan Region; 4—Khabarovsk Territory; 5—Sakhalin Region; 6—Primorye Territory; 7—Jewish Autonomous Region; 8—Amur Region; 9—Republic of Sakha (Yakutia); 10—Trans-Baikal Territory; 11—Republic of Buryatia; 12—Irkutsk Region; 13—Krasnoyarsk Territory; 14—Republic of Tuva; 15—Republic of Khakassia; 16—Republic of Altai; 17—Altai Territory; 18—Kemerovo Region; 19—Novosibirsk Region; 20—Tomsk Region; 21—Omsk Region; 22—Yamal-Nenets Autonomous Area; 23—Khanty-Mansi Autonomous Area–Yugra; 24—Tyumen Region; 25—Sverdlovsk Region; 26—Kurgan Region; 27—Chelyabinsk Region. (Map is based on https://commons.wikimedia.org/wiki/File:Map_of_Russian_subjects,_2008-03-01.svg (accessed on 28 January 2023)).

**Figure 5 plants-12-02792-f005:**
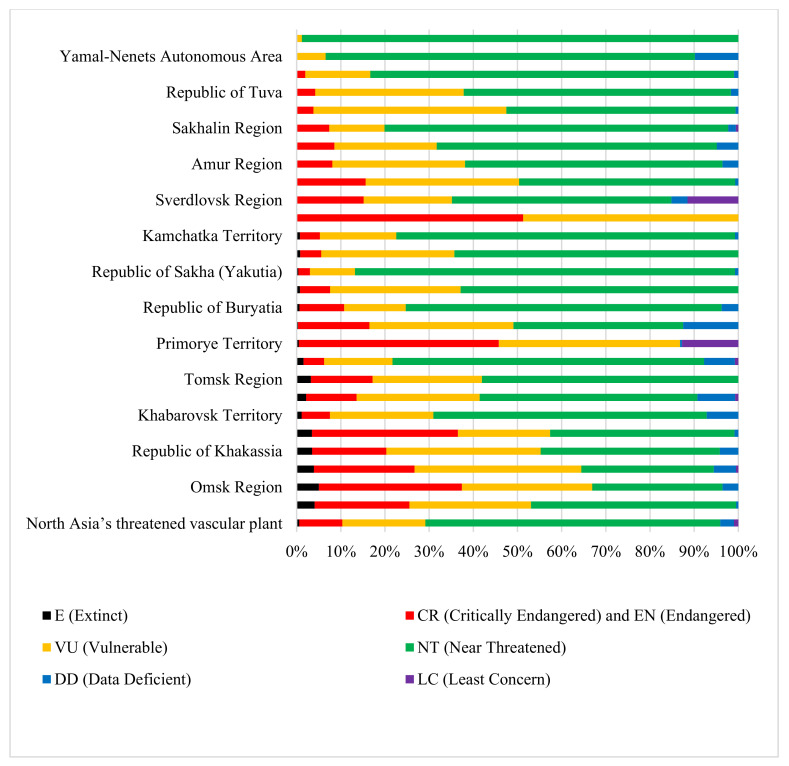
Quantitative statistics for categories of threatened vascular plants in 27 provincial distribution regions of North Asia.

**Figure 6 plants-12-02792-f006:**
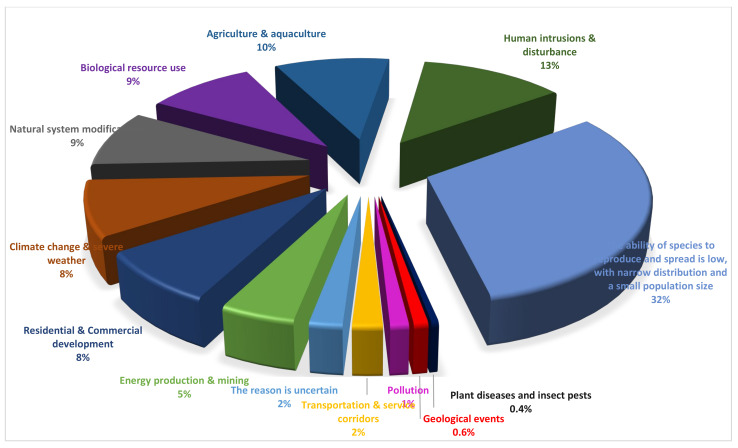
Factors threatening rare and endangered vascular plants in North Asia.

**Table 1 plants-12-02792-t001:** Quantitative statistics of number of rare and endangered vascular plants in North Asia in Red Data Book of the Russian Federation, IUCN Red List for North Asia, and local Red Data Books for 27 provincial administrative regions.

No	Region	Area (km^2^)	Vascular Plants (Angiosperms, Gymnosperms, Ferns) in Red Data Book	Category System (IUCN or Russian)	Vascular Plants **	Reference	Vascular Plants in Red Data Book *	Vascular Plants in Red Data Book of Russian Federation *	Vascular Plants in IUCN Red List for Russia *
1	Kamchatka Territory	464,300	133 (118, 1, 14)	Russian	1075	[10]	133	11	23
2	Chukotka Autonomous Area	721,500	104 (96, 0, 8)	Russian	1008	[11]	103	14	10
3	Magadan Region	462,500	85 (79, 1, 5)	Russian	1364	[12]	85	2	14
4	Khabarovsk Territory	787,600	267 (238, 3, 26)	Russian	2370	[13]	265	51	18
5	Sakhalin Region	87,100	177 (150, 4, 23)	Russian	1858	[14]	176	68	22
6	Primorye Territory	164,700	214 (185, 6, 23)	IUCN	2544	[15]	212	90	19
7	Jewish Autonomous Region	36,300	132 (119, 1, 12)	Russian	1134	[16]	132	28	13
8	Amur Region	361,900	224 (210, 3, 17)	Russian	1960	[17]	223	29	20
9	Republic of Sakha (Yakutia)	3,083,500	266 (249, 3, 14)	Russian	1859	[18]	265	20	32
10	Trans-Baikal Territory	431,500	164 (150, 2, 12)	Russian	1750	[19]	164	18	24
11	Republic of Buryatia	351,300	158 (140, 1, 17)	Russian + IUCN	2042	[20]	158	42	21
12	Irkutsk Region	775,300	180 (171, 0, 9)	Russian	2185	[21]	180	34	27
13	Krasnoyarsk Territory	2,366,600	330 (299, 2, 29)	Russian	2135	[22]	328	37	28
14	Republic of Tuva	168,600	119 (114, 0, 5)	Russian	1952	[23]	119	39	8
15	Republic of Khakassia	61,600	143 (131, 1, 11)	Russian	1570	[24]	143	26	13
16	Republic of Altai	92,900	127 (118, 0, 9)	Russian	2064	[25]	126	36	13
17	Altai Territory	168,000	158 (141, 0, 17)	Russian	2157	[26]	158	29	18
18	Kemerovo Region	95,700	142 (121, 2, 19)	Russian	2250	[27]	141	18	24
19	Novosibirsk Region	177,800	115 (104, 2, 9)	Russian	1286	[28]	115	11	22
20	Tomsk Region	314,400	93 (83, 1, 9)	Russian	1217	[29]	93	10	17
21	Omsk Region	141,100	139 (131, 1, 7)	Russian	1198	[30]	139	14	25
22	Yamal-Nenets Autonomous Area	769,300	61 (58, 0, 3)	Russian	719	[31]	61	4	7
23	Khanty-Mansi Autonomous Area—Yugra	534,800	132 (112, 0, 20)	Russian	1085	[32]	131	9	17
24	Tyumen Region	160,100	140 (127, 0, 13)	Russian	1243	[33]	140	17	20
25	Sverdlovsk Region	194,300	167 (158, 0, 9)	Russian	1532	[34]	165	30	17
26	Kurgan Region	7100	196 (171, 3, 22)	Russian	1045	[35]	196	16	29
27	Chelyabinsk Region	88,500	201 (187, 1, 13)	Russian + IUCN	1959	[36]	197	32	26
	Russian Federation	17,125,191	514 (474, 14, 26)	Russian	-	[6]	511	276	-
	In IUCN	-	908 (835, 57, 16)	IUCN	-	-	908	-	190

* Result of quantitative statistics in this paper. ** Results of this study will be published in a separate article.

## Data Availability

Not applicable.

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
