# Peer review of "Mapping Asia Plants: The Threat Status and Influencing Factors of Rare and Endangered Vascular Plant Species in North Asia (Asian Russia)"

_plants, 2023, doi:10.3390/plants12152792_

Round 1

Reviewer 1 Report

In my opinion, this work does not have a very high scientific value. It is just an attempt to control the chaos that concerns the recognition of plant species in the Asian part of Russia. Of course, the authors have worked hard and it would have to be refined at the level of Russia and published there. I am not sure it is suitable for publication in a journal with a high IF.    

Reviewer 2 Report

This is a very relevant contribution on rare and endangered plants of the Asian part of Russia based on local red books. From my point of view, this contribution includes abundant and interesting information, which is why it deserves to be published in 'Plants'.

For decades, the delimitation of threatened species has been relatively objective since it is based on IUCN criteria, widely accepted and used internationally. However, the delimitation of "rare plants" is highly subjective and problematic given that rarity is a multifaceted concept (demographic, biogeographic? ecological? biological?...). 

The authors adequately address this problem (in the discussion "Mixing the concepts of “rare species” and “endangered species” an also in the Conclusions), but including aspects related to threats and "rarity" in the analysis (Results) may cause  misinterpretations. As, for example, in Lines 275-280: "15 types of factors":

"residential and commercial development, agriculture and aquaculture, energy production and mining, transportation and service corridors, biological resource use, human intrusions and disturbance, natural system modifications, plant diseases and insect pests, geological events, climate change and severe weather, the ability of species to reproduce and spread is low, with narrow distribution, located at the boundary of the distribution area, Relict species, Endemic species."

Indeed there are 15 factors, but the last 2  (Relict species, Endemic species) are not risk factors... but perhaps they could be related to the ambiguous concept of rarity. It would probably be convenient to indicate which of these 15 factors refer to threats and which ones are used to identify "rare plants".

Finally, the discussion and conclusions regarding the limitations of analysis based on data from Regional Red Books and the need to unequivocally identify threatened species without considering "rare plants" are well stated and justified.

Minor comments:

Line 16: "digitalized" instead of "digitized" ?

Line 116 "2. Material and methods" instead of "2. Results"

Line 164: "667 genera" instead of "667genera"

Line 196: "EX (at the federal level)" instead of "E (at the federal level)", since "E" does not exists as a UICN category. This should be corrected in other parts of the text.

Line "100 endemic species of Russian species" rewrite... perhaps "100 species endemic to Russia"?

Lines 346-347: "Hydrocharis morsus-ranae L. and Sagittaria sagittifolia L.": correct font size and italics should be limited to scientific names only

Although I have not focused specifically on the quality of the English, I have detected some errors (mentioned above), so it is advisable to review the text.

Reviewer 3 Report

This is an interesting and potentially valuable manuscript summarising information on rare and threatened vascular plants in the vast area that is Asian Russia. The main conclusion seems to be that the IUCN Red List criteria have not been applied systematically across the different regions, thus limiting the accuracy of the listing status and usefulness for conservation planning and management. However, I have serious concerns about the presentation of the manuscript and found it difficult to follow in many places. The global literature is substantial, including many similar reviews of Red Lists in other countries and regions, yet this is not referenced in the Introduction or Discussion - it really does need to be, to place these results in context. A review of previous global studies could also inform approaches to presenting data, and recommendations for improving the Red List in Asian Russia.

The manuscript is very long, and has a lot of detail around what is included in different red books. There is also quite a lot of repetition in places; e.g. paragraph beginning line 177 is already mostly summarised in the Methods (where it belongs, so remove from Results). Some of the Results could be tabulated or appendicised, to make the paper flow better and highlight the key points.There are too many Figures and some can be easily summarised in a couple of sentences, especially the pie charts (Figures 3 and 4), which I recommend removing and briefly summarising in words.

Grammar and sentence structure are poor in many places throughout the manuscript, thus a thorough English-language edit is recommended prior to publication. Red Book is sometimes capitalised and sometimes not - it should be capitalised throughout the manuscript.

'Statistical analyses' are referred to throughout, but really there haven't been statistics done, it is more that the data have been summarised numerically.

There are many instances where Methods appear in the Results, usually at the first sentence of each paragraph - Methods should clearly state what was done, while Results present the data. Correcting this would remove some of the repetition noted above.

The Abstract needs some editing grammatically and for clarity, e.g. sentence beginning Line 21 is not a proper sentence; the last sentence is also poor.

Headings would help to structure the Introduction, e.g. 'Background to Red Books in the study area', 'Overview of study area'. The place names and regions mentioned in the paragraph beginning line 92 could be shown on a map for an international readership.

Heading 2 on line 116 says 'Results' - should be Methods.

Line 164: don't need to say 'The results of our statistical analysis'. Just start by saying 'We collated a total of 143 families, 667 genera....from the 27 Regional Red Data Books in North Asia'.

Paragraph beginning Line 191 is very long and difficult to follow - I recommend a Table that shows the differences between the IUCN and Russian categories, which could replace this text and perhaps be placed in the Methods.

I recommend the maps in Figures 1,6,7 and 8 be made smaller and placed in a panel of four maps together (thus a single figure with (a-d). This would allow the reader to compare them all at a glance.

Figure 9 - most plants will have more than one threat, so a pie graph is not best to represent these data. I suggest a table would be more informative, and could also include much of the explanatory text in the preceding paragraph, as well as better explaining some of the 'threats' (e.g. why is 'located at the boundary of the distribution area' a threat?)

The Discussion is quite confusing and it is difficult to discern the major points. The Discussion needs to be much shorter and clearer. Red listing should be based on IUCN criteria, thus 'rare' and 'endangered' species should not be confused as there are defined criteria for assessing threat status. Are the authors saying that these have not been applied consistently across the regions?

There is new information presented in the Results, e.g. lines 366-369 on population trends. No new information should appear in the Discussion - this should have been presented in the Results.

There are very few references in the Discussion. The Results of this study need to be placed in the context of global research on threatened species and IUCN Red Lists.

See comments above - needs a thorough edit for clarity and conciseness as noted.

Round 2

Reviewer 3 Report

My comments have generally been addressed. The structure of the manuscript has not been substantially improved or modified, and thus it is still hard to follow and quite confusing in sections. However, in general it is improved and with English language editing should become easier to read. Despite these ongoing concerns, I think it is probably suitable for publication at the Editor's discretion.

see above.

Author Response

We agree with the respected reviewer. We reorganized and adjusted the logical relationship of the full text of the manuscript, removed some redundant content, added several subheadings in the research methods and research results, and strived to make the chapter structure of the full text more reasonable (see revised manuscript).